# THEORETICAL INSIGHTS INTO PSEUDO-LABEL-BASED SEMI-SUPERVISED LEARNING: CONVERGENCE RATE AND SAMPLE COMPLEXITY ANALYSIS

## ABSTRACT

Pseudo-label-based semi-supervised learning has recently emerged as an effective technique in various domains. In this paper, we present a comprehensive theoretical analysis of the algorithm, significantly advancing our understanding of its empirical successes. Our analysis demonstrates that the algorithm can achieve a remarkable convergence rate of $\mathcal{O}(N^{-1/2})$ order, and we provide an estimate of the sample complexity. We further investigate the algorithm's performance in scenarios with an infinite number of unlabeled data points, highlighting its effectiveness in leveraging large-scale unlabeled data. A key insight of our study is that incorporating pseudo-labeled data can improve model training when correctly labeled data is more valuable than the interference caused by mislabeled data, particularly for under-parameterized models that tend to ignore the impact of incorrect labels. Experimental findings corroborate the accuracy of our estimations. This study elucidates the strengths and limitations of the pseudo-label-based semi-supervised learning algorithm, paving the way for future research in this field. The code can be found at the anonymous URL `https://anonymous.4open.science/r/mycode_1-A2EE`.

## 1 INTRODUCTION

Semi-supervised learning, which utilizes both labeled and unlabeled data, has garnered substantial interest in recent years. Among the various methodologies, the pseudo-label-based semi-supervised learning approach has demonstrated its effectiveness in numerous domains. This technique generates pseudo-labels for unlabeled data using a previously trained model, which are then combined with the labeled data to train a new model (Lee, 2013). Recent advancements have seen pseudo-label-based semi-supervised learning achieve unparalleled results across diverse domains, such as (Xie et al., 2020; Guo & Li, 2022; Chen et al., 2020; Kumar et al., 2020; Wang et al., 2022; Sohn et al., 2020; Zhang et al., 2021; Xu et al., 2021; Pham et al., 2021) in the CV field, (Meng et al., 2019; Li et al., 2020; Hsu et al., 2023) in the NLP field, and (Ling et al., 2022; Zia et al., 2022; Dong et al., 2022; Higuchi et al., 2021; Xu et al., 2020; Lugosch et al., 2022) in the speech field.

Despite these empirical triumphs, the theoretical understanding of pseudo-label-based semi-supervised learning is still in its infancy. Earlier correlation analyses, such as those in (Carmon et al., 2019; Raghunathan et al., 2020; Chen et al., 2020; Oymak & Gulcu, 2020), primarily focus on linear models or Gaussian (near-Gaussian) data, which limits their contributions. (Wei et al., 2020) showed that self-training and input-consistency regularization improve accuracy with ground-truth labels. However, they did not address the convergence rate.

In this paper, we aim to bridge this gap by providing a comprehensive theoretical analysis of the pseudo-label-based algorithm. To the best of our knowledge, this is the first study that offers a clear characterization of the convergence rate for general settings without explicit constraints on the network's structure or the data distribution.

A central insight from our research indicates that integrating pseudo-labeled data can bolster model training when the advantages gained from accurately labeled data outweigh the disruptions from inaccurately labeled instances. This is particularly true in cases of *under-parameterized* or when appropriate regularization techniques are applied. Notably, we found that under-parameterized models

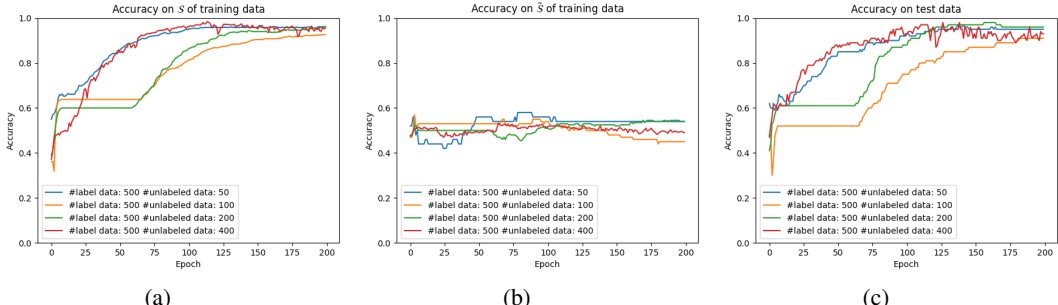

Figure 1: A toy under-parameterized experiment. The data are uniformly sampled within the range $[-1.3, 1.3] \times [-1.3, 1.3]$ in the two-dimensional plane. The ground truth label for each data point is determined as follows: if the point lies inside the unit circle, it is labeled as 1; otherwise, it is labeled as 0. The training set consists of two parts: correctly labeled data, denoted as $\mathcal{S}$, and randomly labeled data, denoted as $\tilde{\mathcal{S}}$. The labels in $\tilde{\mathcal{S}}$ are assigned randomly with an equal probability for labels 0 and 1. We train a three-layer fully connected neural network on the combined set $\mathcal{S} \cup \tilde{\mathcal{S}}$ and evaluate its performance on a separate test set. The size of $\mathcal{S}$ is fixed at $n = 500$, and we conduct experiments with four different sizes of $\tilde{\mathcal{S}}$: $m = 50, 100, 200, 400$. The changes in accuracy on $\mathcal{S}$ are presented in (a), the accuracy on $\tilde{\mathcal{S}}$ in (b), and the accuracy on the test set in (c). We observe that the model tends to disregard the influence of the random data $\tilde{\mathcal{S}}$ during training, as indicated by the accuracy on $\tilde{\mathcal{S}}$ being approximately 0.5. Moreover, the model performs well on the test set across all the different settings.

tend to minimize the effects of incorrect labels. This is exemplified in a simplified scenario depicted in Figure 1, where a model trained with a certain percentage of mislabeled data demonstrates a low empirical error for the correctly labeled instances but shows a comparatively high empirical error for the mislabeled ones. Essentially, the model "sidesteps" the inaccurately labeled data.

In semi-supervised learning situations, it is common to deal with vast amounts of unlabeled data, potentially leading to a plethora of pseudo-labels. This is exactly the scene where our insight above matches. A common scenario is when a model from a prior iteration produces an extensive set of pseudo-labels for an unlabeled dataset. Even though a minor portion of these pseudo-labels might be inaccurately labeled, leveraging these generated pseudo-labels to train an updated model results in the model emphasizing the correct labels and dismissing the inaccuracies. As a result, the subsequent model's performance sees an enhancement.

Our contributions are manifold and are summarized as follows:

- **Estimation of Convergence Rate**: We derive an impressive convergence rate of $\mathcal{O}(N^{-1/2})$ order for the pseudo-label-based semi-supervised learning algorithm. Our analysis sheds light on the factors contributing to its swift convergence, and experimental results corroborate our estimates.

- **Estimation of Sample Complexity**: We provide an estimate of the sample complexity required to achieve the target convergence rate, offering guidelines for practical applications. This insight illuminates the minimum quantity of unlabeled data necessary to reach the desired performance, thus enabling a more efficient utilization of data resources in semi-supervised learning.

- **Investigation of Performance**: We analyze the performance of the algorithm when the number of unlabeled data points is infinite, underscoring its effectiveness in leveraging large-scale unlabeled data. This analysis unveils the potential benefits and limitations of the pseudo-label-based approach under varying data availability scenarios, further emphasizing its practical implications.

The remainder of this paper is organized as follows: Section 2 introduces key preliminaries. Section 3 introduces our key characterization of the model training recipe. Section 4 presents our main results, including optimal population error rate, convergence rate, sample complexity, and perfor-

mance evaluation. Section 5 discusses our experimental setup, datasets, and results. Section 6 reviews the literature on pseudo-label-based semi-supervised learning. Finally, Section 7 concludes our work and discusses future research directions.

## 2 PRELIMINARIES

This section introduces the notations, definitions, and estimation tools that will be used throughout the paper. These preliminaries are essential for understanding the subsequent theoretical analysis of the pseudo-label-based semi-supervised learning algorithm.

### 2.1 NOTATION

We first define the notation that will be consistently used throughout this paper to ensure clarity. The notation is summarized in Table 1.

Table 1: Notation

| Notation | Description | Notation | Description |
|---|---|---|---|
| $k$ | Number of classifications | $\mathcal{E}_{\mathcal{S}}$ | Empirical error rate on $\mathcal{S}$ |
| $a$ | Constant in proposition 4.1 | $\mathcal{E}_{\tilde{\mathcal{S}}}$ | Empirical error rate on $\tilde{\mathcal{S}}$ |
| $\mathcal{S}$ | Correctly labeled data | $\mathcal{E}_{\mathcal{D}}$ | Population error rate |
| $\tilde{\mathcal{S}}$ | Randomly labeled data | $\mathcal{E}_D^*$ | Optimal population error rate |
| $p^*$ | Target convergence rate | $m$ | Number of randomly labeled data |
| $n$ | Number of correctly labeled data | $N$ | Total number of data ($N = m + n$) |
| $f_0$ | Pseudo labeler | $f_1$ | Output model |
| $\varepsilon, \tilde{\delta}, b$ | Parameters in Definition 3.1 | | |

### 2.2 POPULATION ERROR ESTIMATION

Several approaches have been proposed to estimate the population risk associated with Deep Neural Network (DNN) models. The most prevalent method involves calculating an upper bound for the population error by considering the complexity of the hypothesis classes (Bartlett & Mendelson, 2002; Shalev-Shwartz & Ben-David, 2014; Hardt et al., 2016; Mukherjee et al., 2006; Allen-Zhu et al., 2019; Neyshabur et al., 2015; 2017; Ma et al., 2018; Weinan et al., 2019). However, this technique is typically customized to the specific model under examination. A new perspective in this field was brought forth by (Garg et al., 2021).

## 3 QUANTIFYING MODEL BEHAVIOR

To conduct a comprehensive analysis, beyond just focusing on specific network architectures or data distributions, it's essential to have a general characterization of the training behavior of neural network models. This is particularly crucial when considering the behavior of neural networks trained on data that is mixed with noisy instances. In this subsection, we provide a mathematical formulation that quantitatively captures the behavior of a model when trained on a dataset consisting of both correctly labeled and randomly labeled data. This formulation primarily delves into the model's sensitivity to the randomly labeled training data. A higher accuracy on this subset implies a greater vulnerability of the model to disturbances caused by these data instances.

It's well-understood that a complete training process comprises three core components: the model, the data, and the optimization. Hence, we refer to a specific combination of model, data, and optimizer as a "training recipe." Using this training recipe, we can derive the model. Essentially, our focus is on characterizing this training recipe. We evaluate a training recipe applied to $N$ data points. Out of these $N$ data points, $n$ are correctly labeled ($\mathcal{S}$), and $m$ are randomly labeled ($\tilde{\mathcal{S}}$).

**Definition 3.1.** $(N - (\varepsilon, \tilde{\delta}, b)$ ***training recipe***$)$ *A training recipe is defined as a $N - (\varepsilon, \tilde{\delta}, b)$ recipe if we have:*

$$\frac{m}{n} \leq \tilde{\delta} < 1, m + n = N \tag{1}$$

*Then we can procure a model $\hat{f}$ by the recipe that fulfills:*

$$\mathcal{E}_{\mathcal{S}}(\hat{f}) \leq \varepsilon \tag{2}$$

$$\mathcal{E}_{\tilde{\mathcal{S}}}(\hat{f}) \geq 1 - \frac{1 + b\varepsilon}{k} \tag{3}$$

In this definition, $\tilde{\delta}$ is a positive parameter representing the ratio of randomly labeled data $(m)$ to the correct data $(n)$, subject to the condition $\frac{m}{n} \leq \tilde{\delta} < 1$. $\varepsilon$ is a positive parameter signifying the model's error rate on the correctly labeled dataset $\mathcal{S}$, and $b$ is a non-negative parameter signifying the error rate on the randomly labeled dataset $\tilde{\mathcal{S}}$. This is indicative of the model's sensitivity to disturbances induced by the randomly labeled data.

## 4 MAIN RESULTS

This section presents a comprehensive theoretical exploration of the pseudo-label-based algorithm for semi-supervised learning. Our analysis covers the optimal population error analysis, convergence rate estimation, sample complexity analysis, and the algorithm's performance analysis under infinite unlabeled data.

### 4.1 OPTIMAL POPULATION ERROR

Our analysis commences by exploring the optimal population error associated with the pseudo-label-based algorithm, which is vital for comprehending its capabilities and limitations. Initially, drawing from (Garg et al., 2021), we present a corollary that offers a more streamlined framework for our examination.

**Proposition 4.1.** *For a $\hat{f}$ trained by $N - (\varepsilon, \tilde{\delta}, b)$ training recipe, then with probability at least $1 - \delta$, $\hat{f}$ satisfies*

$$\mathcal{E}_{\mathcal{D}}(\hat{f}) \leq \mathcal{E}_{\mathcal{S}}(\hat{f}) + (k - 1)\left(1 - \frac{k}{k - 1}\mathcal{E}_{\tilde{\mathcal{S}}}(\hat{f})\right) + ak\sqrt{\frac{\log\left(\frac{4}{\delta}\right)}{m}} \tag{4}$$

*where $a$ is a constant and satisfies $a < 4$.*

*Furthermore, if the $\tilde{\delta}$ satisfies*

$$2k + \sqrt{k} + \frac{\tilde{\delta}}{\sqrt{k}} < 2\sqrt{2}k \tag{5}$$

*then we have*

$$\mathcal{E}_{\mathcal{D}}(\hat{f}) \leq \mathcal{E}_{\mathcal{S}}(\hat{f}) + (k - 1)\left(1 - \frac{k}{k - 1}\mathcal{E}_{\tilde{\mathcal{S}}}(\hat{f})\right) + 2k\sqrt{\frac{\log\left(\frac{4}{\delta}\right)}{m}} \tag{6}$$

We denote the optimal population error of model $\hat{f}$, trained by an $N - (\varepsilon, \tilde{\delta}, b)$ training recipe, as $\mathcal{E}_D^*$. We employ proposition 4.1 to derive the optimal population error, $\mathcal{E}_D^*$. The term $\mathcal{E}_D^*$ quantifies the population error when the model $\hat{f}$ is trained on $N$ correctly labeled data points. We have:

$$\mathcal{E}_D^* = (1 + b)\varepsilon + ak\sqrt{\log\left(\frac{4}{\delta}\right)}\frac{1}{\sqrt{\frac{\tilde{\delta}}{1 + \tilde{\delta}}N}} \tag{7}$$

*Proof sketch*: We utilize proposition 4.1 to estimate $\mathcal{E}_D^*$. A significant advantage of this approach is its applicability to the general model. Note that our model, $\hat{f}$, is trained by an $N - (\varepsilon, \tilde{\delta}, b)$ training

recipe, necessitating $\mathcal{E}_{\mathcal{S}}$, $\mathcal{E}_{\tilde{\mathcal{S}}}$, and the permissible relative proportion of randomly labeled data, $\tilde{\delta}$. Therefore, on the $N$ data points that are all correctly labeled, we can select at most $m = \frac{\tilde{\delta}}{1+\tilde{\delta}} N$ data points as randomly labeled to estimate the model's population error. At this point, the right-hand side of inequality 4 reaches its minimum, providing the optimal population error estimate of $\hat{f}$. More detailed derivations can be found in the Appendix.

**Remark 4.2.** *We can also observe that $\mathcal{E}_D^*$ is optimal from another perspective, as it has the error rate order of $O(\frac{1}{\sqrt{N}})$, which is equal to the error rate order of Monte Carlo estimation. This observation implies the reasonableness of our definition 3.1.*

## 4.2 CONVERGENCE RATE ESTIMATION

In this section, we estimate the convergence rate of the pseudo-label-based algorithm. The convergence rate is critical for evaluating the algorithm's efficiency and effectiveness throughout the learning process. Specifically, we consider an iteration where $f_0$ is the preceding model, and $f_1$ is the current model to be trained. Pseudo labels are generated by $f_0$ and used to train $f_1$. This convergence rate estimation enables us to assess the efficacy of the method. Our main conclusion is as follows:

**Theorem 4.3** (**Convergence Rate Estimation**). *For $f_1$ trained by $N - (\varepsilon, \tilde{\delta}, b)$ training recipe, if*

$$\mathcal{E}_D(f_0) \leq \frac{\tilde{\delta}}{1+\tilde{\delta}} \tag{8}$$

*then with at least $(1 - \delta)^2$ probability, we have*

$$p \leq \frac{ak}{(\mathcal{E}_D(f_0) - \mathcal{E}_D^*)\sqrt{N}} \left[ \sqrt{\frac{\tilde{\delta} + 1}{\tilde{\delta} - \frac{\mathcal{E}_D(f_0)}{1 - \mathcal{E}_D(f_0)}}} - \sqrt{\frac{\tilde{\delta} + 1}{\tilde{\delta}}} \right] \cdot \sqrt{\log\left(\frac{4}{\delta}\right)} \tag{9}$$

*Here, $a$ is the constant in proposition 4.1 and the definition of the convergence rate $p$ is:*

$$p \triangleq \frac{\mathcal{E}_D(f_1) - \mathcal{E}_D^*}{\mathcal{E}_D(f_0) - \mathcal{E}_D^*} \tag{10}$$

*Proof Sketch*: For the pseudo-labels generated by $f_0$, we denote the population loss of the $f_0$ model as $\mathcal{E}_D(f_0)$. This implies that among all $N$ data points, there are approximately $(1 - \mathcal{E}_D(f_0))N$ correctly labeled instances and $\mathcal{E}_D(f_0)N$ wrongly labeled instances. However, we cannot treat the wrongly labeled instances in the generated pseudo-labels as random labels, since there may be some structure in the wrong labels that prevents these errors from being uniformly distributed. To circumvent this issue, in our analysis, we approximate the original algorithm by randomly selecting a small number of pseudo-labels and re-randomizing their labels. We then use proposition 4.1 to estimate the convergence rate. This approximation is akin to analyzing a slightly weakened version of the pseudo-label semi-supervised learning algorithm, which further ensures the robustness of our convergence rate estimation for the pseudo-label semi-supervised learning algorithm. Since the $f_1$ is trained by the $N - (\varepsilon, \tilde{\delta}, b)$ recipe, we can select up to $\frac{\tilde{\delta}(1-\mathcal{E}_D(f_0)) - \mathcal{E}_D(f_0)}{(1+\tilde{\delta})(1-\mathcal{E}_D(f_0))} N$ samples from the pseudo-labels to re-randomize.

Compared with the setting in Subsection 4.1, where we have all correct labels, the allowable number of re-randomized labels here is reduced due to the errors in pseudo-labels. A higher $\mathcal{E}_D(f_0)$ will reduce the allowable number of re-randomized labels, thereby leading to a higher population error. We can further derive that the population error of $f_1$ satisfies:

$$\mathcal{E}_D(f_1) \leq (1+b)\varepsilon + ak\sqrt{\log\left(\frac{4}{\delta}\right)} \sqrt{\frac{(1+\tilde{\delta})(1-\mathcal{E}_D(f_0))}{\tilde{\delta}(1-\mathcal{E}_D(f_0)) - \mathcal{E}_D(f_0)}} \frac{1}{\sqrt{N}} \triangleq upper(f_1) \tag{11}$$

By estimating the numerator of equation 10, we arrive at our desired Theorem 4.3. This theorem provides two significant insights into the convergence rate of the pseudo-label-based algorithm:

- The convergence rate is of order $\mathcal{O}(N^{-\frac{1}{2}})$. This suggests that as the number of unlabeled data, denoted as $N$, increases, the convergence rate improves, demonstrating a reciprocal square-root relationship. This implies that the greater the quantity of unlabeled data available, the quicker we can anticipate our algorithm to converge to a solution. This is a highly desirable property, especially in scenarios where large volumes of unlabeled data are readily accessible.
- The term $\mathcal{E}_D(f_0) - \mathcal{E}_D^*$ in the denominator indicates that as the population loss of the preceding model, denoted as $\mathcal{E}_D(f_0)$, approaches the optimal population error $\mathcal{E}_D^*$, the convergence rate begins to decay. This means that the convergence rate is high when $\mathcal{E}_D(f_0)$ is relatively large, that is, when the preceding model is far from optimal. As the model improves and $\mathcal{E}_D(f_0)$ approaches $\mathcal{E}_D^*$, the rate of improvement decelerates. This observation aligns with practical experiences in pseudo-label applications, where the boost will gradually decrease.

## 4.3 SAMPLE COMPLEXITY ESTIMATION

In this section, we explore the sample complexity of the pseudo-label-based algorithm. This analysis is vital for determining the minimum amount of unlabeled data required to achieve the desired convergence rate. We outline the conditions on the number of unlabeled samples, $N$, necessary to ensure that the convergence rate remains below the target rate, $p^*$, with a high probability. Our primary result is presented as follows:

**Theorem 4.4.** *For $f_1$ trained by $N - (\varepsilon, \tilde{\delta}, b)$ training recipe, with at least $(1 - \delta)^2$ probability, we have*

$$p \triangleq \frac{\mathcal{E}_D(f_1) - \mathcal{E}_D^*}{\mathcal{E}_D(f_0) - \mathcal{E}_D^*} \leq p^* \tag{12}$$

*when*

$$N \geq \left(\frac{ak}{p^*(\mathcal{E}_D(f_0) - \mathcal{E}_D^*)}\right)^2 \left[\sqrt{\frac{\tilde{\delta} + 1}{\tilde{\delta} - \frac{\mathcal{E}_D(f_0)}{1 - \mathcal{E}_D(f_0)}}} - \sqrt{\frac{\tilde{\delta} + 1}{\tilde{\delta}}}\right]^2 \cdot \log\left(\frac{4}{\delta}\right) \tag{13}$$

*and*

$$\mathcal{E}_D(f_0) \leq \frac{\tilde{\delta}}{1 + \tilde{\delta}} \tag{14}$$

*Proof Sketch*: The objective is to achieve the target convergence rate $p^*$ by ensuring the following inequality holds:
$$\frac{\mathcal{E}_D(f_1) - \mathcal{E}_D^*}{\mathcal{E}_D(f_0) - \mathcal{E}_D^*} \triangleq p \leq p^* \tag{15}$$

We can derive:
$$p \triangleq \frac{\mathcal{E}_D(f_1) - \mathcal{E}_D^*}{\mathcal{E}_D(f_0) - \mathcal{E}_D^*} \leq \frac{upper(f_1) - \mathcal{E}_D^*}{\mathcal{E}_D(f_0) - \mathcal{E}_D^*} \triangleq \tilde{p} \tag{16}$$

Therefore, to ensure that inequality 15 holds, the following condition must be satisfied:
$$\tilde{p} \leq p^* \tag{17}$$

By solving inequality equation 17, we reach the desired conclusion. More detailed reasoning is provided in the Appendix.

As outlined in our theorem 4.4, there exists an inverse square relationship between the sample complexity and the target convergence rate. Specifically, as the target convergence rate $p^*$ decreases, the requisite number of data $N$ increases. This relationship can be intuitively understood as follows: aiming for a faster convergence rate $p^*$ inherently necessitates a larger number of pseudo samples $N$. This is due to the fact that more pseudo samples offer more information for the algorithm to learn and adjust its parameters, thereby accelerating its convergence to the optimal solution. This understanding is vital for the design and implementation of pseudo-label-based algorithms as it enables

researchers and practitioners to balance the trade-off between the desired speed of convergence and the available pseudo-data resources. The sample complexity analysis enhances our understanding of the algorithm's data requirements. We further explore the algorithm's performance when the number of unlabeled data points is infinite, shedding light on its capabilities.

## 4.4 ANALYSIS OF THE INFINITE UNLABELED DATA SCENARIO

In this section, we investigate the behavior of the pseudo-label-based semi-supervised learning algorithm when an infinite number of unlabeled data points are present. By analyzing the algorithm's performance under this limitation, we aim to underscore the strengths of the method.

Firstly, from equation 7, we have:

$$\mathcal{E}_D^* \triangleq (1+b)\varepsilon + ak\sqrt{\log\left(\frac{4}{\delta}\right)}\frac{1}{\sqrt{\frac{\tilde{\delta}}{1+\tilde{\delta}}N}} \tag{18}$$

And for $f_1$ trained by $N - (\varepsilon, \tilde{\delta}, b)$ training recipe, with a probability of at least $1 - \delta$, we have 11:

$$\mathcal{E}_D(f_1) \leq (1+b)\varepsilon + ak\sqrt{\log\left(\frac{4}{\delta}\right)}\sqrt{\frac{(1+\tilde{\delta})(1-\mathcal{E}_D(f_0))}{\tilde{\delta}(1-\mathcal{E}_D(f_0))-\mathcal{E}_D(f_0)}}\frac{1}{\sqrt{N}} \tag{19}$$

As $N$ approaches infinity, we can establish the following relationship between the population risk of $f_1$ and the optimal population risk $\mathcal{E}_D^*$:

$$\lim_{N\to+\infty}\frac{\mathcal{E}_D(f_1)}{\mathcal{E}_D^*} = 1 \tag{20}$$

Given these findings, we can now present the following theorem:

**Theorem 4.5.** *For $f_1$ trained by $N - (\varepsilon, \tilde{\delta}, b)$ training recipe, if $\mathcal{E}_D(f_0) < \frac{\tilde{\delta}}{1+\tilde{\delta}}$ then with at least $1 - \delta$ probability, we have:*

$$\lim_{N\to+\infty}\frac{\mathcal{E}_D(f_1)}{\mathcal{E}_D^*} = 1 \tag{21}$$

This result has significant implications for the pseudo-label-based semi-supervised learning algorithm. It demonstrates that as the number of unlabeled data points increases, the performance of the algorithm converges to the optimal population error in one iteration, thus showcasing the effectiveness of the algorithm when leveraging a large amount of unlabeled data. The examination of the infinite unlabeled data scenario underscores the algorithm's adaptability under various conditions.

## 5 NUMERICAL EXPERIMENTS

### 5.1 EXPERIMENTAL SETUP

In this section, we describe the experimental setup designed to validate our theoretical findings, particularly on the convergence rate estimation. Despite the inherent challenges, experiments are conducted on real-world datasets using the widely-adopted ResNet architecture He et al. (2016). To ensure that the convergence rate estimation in Theorem 4.3 is less than 1 (thus making it meaningful), additional modifications are made to the CIFAR-10 and FashionMNIST datasets:

(a) The 10-class classification problem is transformed into a 2-class problem. Specifically, for CIFAR-10, images of airplanes, cars, birds, cats, and deer are labeled as 0, whereas images of dogs, frogs, horses, ships, and trucks are labeled as 1. For FashionMNIST, T-shirt, trouser, pullover, dress, and coat images are labeled as 0, while sandal, shirt, sneaker, bag, and ankle boot images are labeled as 1.

(b) Data augmentation techniques, such as random horizontal flip and random crop, are employed to obtain more samples. Specifically, 240,000 samples (four times the original dataset size) from CIFAR-10 and 180,000 samples (three times the original dataset size) from FashionMNIST are generated using data augmentation.

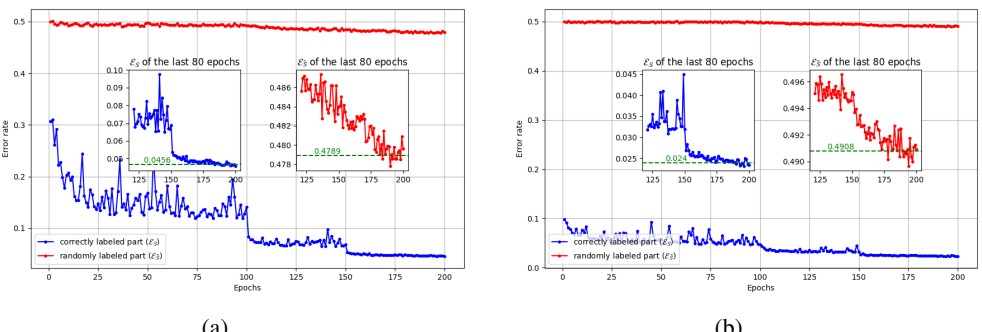



(a)                      (b)



Figure 2: Estimation of $\varepsilon$ and $b$ for the training recipe on (a) CIFAR-10 and (b) FashionMNIST datasets. The blue line represents the model's error rate on the correctly labeled part of the training set ($\mathcal{E}_{\mathcal{S}}$), and the red line represents the model's error rate on the randomly labeled part of the training set ($\mathcal{E}_{\tilde{\mathcal{S}}}$).

The training recipe employs a ResNet model composed of three layers of BasicBlock units with output channels of 16, 32, and 64, subsequently succeeded by a final linear projection layer. Each BasicBlock features two Conv2d layers with 3x3 kernels, trailed by BatchNorm2d layers. The optimization hyperparameters are established as follows: a learning rate (LR) of 0.1, momentum of 0.9, weight decay of 1e-4, and a mini-batch size set to 128. The model undergoes training for 200 epochs on two 3090 GPUs over a few hours. Refer to the Appendix for more detailed information.

## 5.2 Validation of convergence rate estimation

This section presents the results of the experiments, which validate the convergence rate estimation. First, the $N - (\varepsilon, \tilde{\delta}, b)$ for the training recipe on the two datasets needs to be estimated respectively. A strategy is adopted in which models are trained on $N$ data, including a $\tilde{\delta}$ proportion of randomly labeled data $\tilde{\mathcal{S}}$, and then the error rate $\mathcal{E}_{\mathcal{S}}$ on the correctly labeled part of the training set and the error rate $\mathcal{E}_{\tilde{\mathcal{S}}}$ on the randomly labeled part are measured, thereby estimating $\varepsilon$ and $b$. The experimental results are shown in Figure 2. The blue line represents the model's $\mathcal{E}_{\mathcal{S}}$, and the red line represents the model's $\mathcal{E}_{\tilde{\mathcal{S}}}$. In both experiments, $\mathcal{E}_{\tilde{\mathcal{S}}}$ remains close to 0.5, indicating that this part of the data has a minor impact on the model's final generalization error.

For the experiment on the CIFAR-10 dataset, we set $\tilde{\delta} = 0.3$, and after training, we measured $\mathcal{E}_{\mathcal{S}} = 0.0456$ and $\mathcal{E}_{\tilde{\mathcal{S}}} = 0.4789$, as shown in Figure 2a. Thus, we obtained suitable $\varepsilon = 0.0465$, and $b = 0.906$. For the experiment on the FashionMNIST dataset, we set $\tilde{\delta} = 0.35$, and after training, we measured $\mathcal{E}_{\mathcal{S}} = 0.024$ and $\mathcal{E}_{\tilde{\mathcal{S}}} = 0.4908$, as shown in Figure 2b. Thus, we obtained suitable $\varepsilon = 0.024$, and $b = 0.77$.

We obtained $f_0$ with a test error of 0.183 on CIFAR-10 and a test error rate of 0.095 on FashionMNIST, and set $\delta$ to 0.05 as a small probability for estimating the convergence rate by Theorem 4.3. To provide a more intuitive demonstration, we converted the estimated convergence rates into an estimation of the population error (test error) for the obtained model $f_1$. The estimations indicated that the estimated population error for $f_1$ on CIFAR-10 should be lower than 0.166 (the green line in Subfigure 3a), and on FashionMNIST, it should be lower than 0.091 (the green line in Subfigure 3b). These estimates align well with the actual experimental results, as demonstrated in Figure 3.

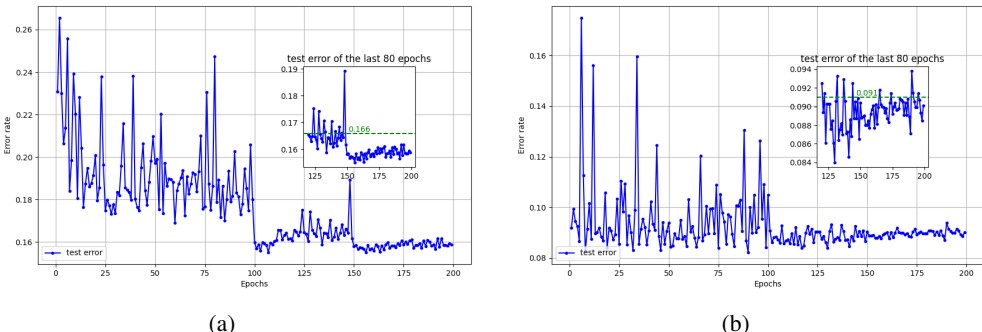

(a)                                               (b)

Figure 3: Comparison of Estimated and Actual Test Errors: (a) CIFAR-10 Experiment; (b) Fashion-MNIST Experiment. The green line represents the estimated error. It is observable that the estimates on both datasets align well with the actual experimental results.

## 6 RELATED WORK

In this section, a concise overview of the theoretical analysis literature in the field of semi-supervised learning algorithms is provided. The effectiveness of pseudo label, a simple yet potent semi-supervised learning method, in achieving robust accuracy levels comparable to the high standard accuracy obtained using a similar quantity of labeled examples, was highlighted by the work of Carmon et al. (2019). Chen et al. (2020) demonstrated that the reliance on spurious features can be inhibited by using entropy minimization on unlabeled target data. Oymak & Gulcu (2020) underscored the advantages of excluding low-confidence samples, showing that iterations of self-training can improve model accuracy, even when trapped in sub-optimal fixed points. However, the analyses presented in Carmon et al. (2019), Raghunathan et al. (2020), Chen et al. (2020), and Oymak & Gulcu (2020) primarily focus on linear models or data that is Gaussian or near-Gaussian.

More recently, Wei et al. (2020) established that self-training and input-consistency regularization can achieve high accuracy relative to ground-truth labels, deriving sample complexity guarantees for neural networks. Nevertheless, the convergence rate was not discussed in Wei et al. (2020). Despite the valuable insights offered by these studies, there are still areas left unaddressed in the theoretical analysis of pseudo-label-based semi-supervised learning algorithms, especially concerning the convergence rate and the applicability to various neural network architectures. In contrast, our study presents a comprehensive theoretical analysis of the pseudo-label-based semi-supervised learning algorithm, filling the gaps in the current body of literature, specifically concerning the convergence rate and the applicability to a wide variety of neural network models.

## 7 CONCLUSION, LIMITATIONS, AND FUTURE WORK

In this paper, we have made significant contributions to understanding the convergence rates and sample complexity of pseudo-label-based semi-supervised algorithms. Our analysis demonstrates that the algorithms can attain an impressive convergence rate of $\mathcal{O}(N^{-1/2})$ order. Experimental results corroborate the accuracy of our estimation. Furthermore, we provide an estimate of the sample complexity and assess the algorithms' performance with an infinite amount of unlabeled data, showcasing its effectiveness in leveraging large quantities of unlabeled data. We posit that our work establishes a solid foundation for future studies in this field.

However, our current work has certain limitations. A primary challenge lies in the efficient and cost-effective estimation of the $N - (\varepsilon, \tilde{\delta}, b)$ parameters that dictate the behavior of algorithms. The estimation of these parameters is vital as they directly influence the estimation of the convergence rate and sample complexity. In our future work, we aim to undertake a more extensive analysis, specifically focusing on the relationship between $(\varepsilon, \tilde{\delta}, b)$ and $N$, which will enable us to devise strategies for estimating these parameters more effectively. This endeavor will further augment the significance of our work.

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

# A APPENDIX A: SUPPLEMENTARY PROOFS

To facilitate the understanding of this appendix, we restate Proposition 3.1.

**Corollary A.1.** *Suppose $\hat{f}$ is trained using the $N - (\varepsilon, \tilde{\delta}, b)$ training recipe. Then, with a probability of at least $1 - \delta$, $\hat{f}$ satisfies the following inequality:*

$$\mathcal{E}_{\mathcal{D}}(\hat{f}) \leq \mathcal{E}_{\mathcal{S}}(\hat{f}) + (k-1)\left(1 - \frac{k}{k-1}\mathcal{E}_{\tilde{\mathcal{S}}}(\hat{f})\right) + ak\sqrt{\frac{\log\left(\frac{4}{\delta}\right)}{m}} \tag{22}$$

*where $a < 4$ is a constant.*

*Additionally, if $\tilde{\delta}$ satisfies*

$$2k + \sqrt{k} + \frac{\tilde{\delta}}{\sqrt{k}} < 2\sqrt{2}k \tag{23}$$

*then we have*

$$\mathcal{E}_{\mathcal{D}}(\hat{f}) \leq \mathcal{E}_{\mathcal{S}}(\hat{f}) + (k-1)\left(1 - \frac{k}{k-1}\mathcal{E}_{\tilde{\mathcal{S}}}(\hat{f})\right) + 2k\sqrt{\frac{\log\left(\frac{4}{\delta}\right)}{m}} \tag{24}$$

**Optimal Population Error** We denote the population error when the model $\hat{f}$ is trained by an $N - (\varepsilon, \tilde{\delta}, b)$ training recipe on $N$ correctly labeled data points as $\mathcal{E}_D^*$. This represents the optimal population error of the pseudo-label semi-supervised learning algorithm. Using Corollary A.1, we derive:

$$\mathcal{E}_D^* = (1 + b)\varepsilon + ak\sqrt{\log\left(\frac{4}{\delta}\right)}\frac{1}{\sqrt{\frac{\tilde{\delta}}{1+\tilde{\delta}}N}} \tag{25}$$

*Proof.* Training $\hat{f}$ with an $N - (\varepsilon, \tilde{\delta}, b)$ recipe accounts for $\mathcal{E}_{\mathcal{S}}$, $\mathcal{E}_{\tilde{\mathcal{S}}}$, and $\tilde{\delta}$, which is the allowed proportion of randomly labeled data. The second term in Corollary A.1 is related to the performance on randomly labeled data $\tilde{\mathcal{S}}$. The third term, $ak\sqrt{\frac{\log\left(\frac{4}{\delta}\right)}{m}}$, decreases as $m$ increases.

For an $N - (\varepsilon, \tilde{\delta}, b)$ training recipe, the maximum $m$ satisfies

$$\begin{aligned}\frac{m}{n} &= \tilde{\delta}, \\ m + n &= N,\end{aligned} \tag{26}$$

which yields

$$\begin{aligned}m &= \frac{\tilde{\delta}}{1 + \tilde{\delta}}N, \\ n &= \frac{1}{1 + \tilde{\delta}}N.\end{aligned} \tag{27}$$

Subsequently, we obtain

$$\mathcal{E}_D(\hat{f}) \leq (1 + b)\varepsilon + ak\sqrt{\log\left(\frac{4}{\delta}\right)}\frac{1}{\sqrt{\frac{\tilde{\delta}}{1+\tilde{\delta}}N}} \tag{28}$$

The right-hand side of the equation is $\mathcal{E}_D^*$. □

**Estimation of Convergence Rate** We proceed to prove our result concerning the estimation of the convergence rate:

**Theorem A.2 (Estimation of Convergence Rate).** *For a model $f_1$ trained utilizing the $N - (\varepsilon, \tilde{\delta}, b)$ training recipe, if*

$$\mathcal{E}_D(f_0) \leq \frac{\tilde{\delta}}{1 + \tilde{\delta}}, \tag{29}$$

*then, with a probability of at least $(1 - \delta)^2$, we have*

$$p \leq \frac{ak}{(\mathcal{E}_D(f_0) - \mathcal{E}_D^*)\sqrt{N}} \left[ \sqrt{\frac{\tilde{\delta} + 1}{\tilde{\delta} - \frac{\mathcal{E}_D(f_0)}{1 - \mathcal{E}_D(f_0)}}} - \sqrt{\frac{\tilde{\delta} + 1}{\tilde{\delta}}} \right] \cdot \sqrt{\log\left(\frac{4}{\delta}\right)}, \tag{30}$$

*where $a$ is the constant from proposition 4.1, and the convergence rate $p$ is defined as:*

$$p \triangleq \frac{\mathcal{E}_D(f_1) - \mathcal{E}_D^*}{\mathcal{E}_D(f_0) - \mathcal{E}_D^*}. \tag{31}$$

Let $\mathcal{E}_D(f_0)$ denote the population error of model $f_0$. This means there are approximately $(1 - \mathcal{E}_D(f_0))N$ correctly labeled instances and $\mathcal{E}_D(f_0)N$ incorrectly labeled instances out of all $N$ data points. However, these incorrectly labeled instances are not random labels as they might arise from structured errors.

To tackle this, we approximate the original algorithm in our analysis by choosing a small fraction of pseudo-labels randomly and re-randomizing their labels. This approach allows us to analyze a slightly weakened version of the pseudo-label semi-supervised learning algorithm, enabling us to use proposition 4.1 to estimate the convergence rate, as well as strengthen our convergence rate estimation.

*Proof.* Given a model $f_1$ trained using the $N - (\varepsilon, \tilde{\delta}, b)$ recipe, the quantity of data selected for re-random labeling, $m$, satisfies

$$\frac{m + \mathcal{E}_D(f_0)(N - m)}{(1 - \mathcal{E}_D(f_0))(N - m)} \leq \tilde{\delta}, \tag{32}$$

which gives

$$m \leq \frac{\tilde{\delta}(1 - \mathcal{E}_D(f_0)) - \mathcal{E}_D(f_0)}{(1 + \tilde{\delta})(1 - \mathcal{E}_D(f_0))} N, \tag{33}$$

and

$$\mathcal{E}_D(f_0) \leq \frac{\tilde{\delta}}{1 + \tilde{\delta}}. \tag{34}$$

Equation 33 implies that at most $\frac{\tilde{\delta}(1 - \mathcal{E}_D(f_0)) - \mathcal{E}_D(f_0)}{(1 + \tilde{\delta})(1 - \mathcal{E}_D(f_0))} N$ data can be chosen from the $N$ generated pseudo-labels for re-random labeling. Next, according to Corollary A.1, we train $f_1$ with at least a $1 - \delta$ probability as

$$\mathcal{E}_D(f_1) \leq (1 + b)\varepsilon + ak\sqrt{\log\left(\frac{4}{\delta}\right)} \sqrt{\frac{(1 + \tilde{\delta})(1 - \mathcal{E}_D(f_0))}{\tilde{\delta}(1 - \mathcal{E}_D(f_0)) - \mathcal{E}_D(f_0)}} \frac{1}{\sqrt{N}}. \tag{35}$$

We define

$$upper(f_1) \triangleq (1 + b)\varepsilon + ak\sqrt{\log\left(\frac{4}{\delta}\right)} \sqrt{\frac{(1 + \tilde{\delta})(1 - \mathcal{E}_D(f_0))}{\tilde{\delta}(1 - \mathcal{E}_D(f_0)) - \mathcal{E}_D(f_0)}} \frac{1}{\sqrt{N}}, \tag{36}$$

and we have

$$p \triangleq \frac{\mathcal{E}_D(f_1) - \mathcal{E}_D^*}{\mathcal{E}_D(f_0) - \mathcal{E}_D^*} \leq \frac{upper(f_1) - \mathcal{E}_D^*}{\mathcal{E}_D(f_0) - \mathcal{E}_D^*}. \tag{37}$$

Let $C \triangleq \mathcal{E}_D(f_0) - \mathcal{E}_D^*$ to simplify the expression, we then obtain:

$$p \leq \frac{upper(f_1) - \mathcal{E}_D^*}{C}. \tag{38}$$

Substituting $upper(f_1)$ and $\mathcal{E}_D^*$ from Equations 36 and 25, and rearranging terms, we get:

$$p \leq \frac{ak\sqrt{\log\left(\frac{4}{\delta}\right)}\left[\sqrt{\frac{(1+\tilde{\delta})(1-\mathcal{E}_D(f_0))}{\tilde{\delta}(1-\mathcal{E}_D(f_0))-\mathcal{E}_D(f_0)}} - \sqrt{\frac{\tilde{\delta}}{1+\tilde{\delta}}}\right]}{C\sqrt{N}}. \tag{39}$$

We can rewrite the term in the square root on the right side of equation 39 as:

$$\sqrt{\frac{(1+\tilde{\delta})(1-\mathcal{E}_D(f_0))}{\tilde{\delta}(1-\mathcal{E}_D(f_0))-\mathcal{E}_D(f_0)}} = \sqrt{\frac{\tilde{\delta}+1}{\tilde{\delta}-\frac{\mathcal{E}_D(f_0)}{1-\mathcal{E}_D(f_0)}}}, \tag{40}$$

Substituting equation 40 into equation 39, the inequality simplifies to:

$$p \leq \frac{ak\sqrt{\log\left(\frac{4}{\delta}\right)}\left[\sqrt{\frac{\tilde{\delta}+1}{\tilde{\delta}-\frac{\mathcal{E}_D(f_0)}{1-\mathcal{E}_D(f_0)}}} - \sqrt{\frac{\tilde{\delta}+1}{\tilde{\delta}}}\right]}{C\sqrt{N}}, \tag{41}$$

Further simplification yields

$$p \leq \frac{ak}{(\mathcal{E}_D(f_0) - \mathcal{E}_D^*)\sqrt{N}}\left[\sqrt{\frac{\tilde{\delta}+1}{\tilde{\delta}-\frac{\mathcal{E}_D(f_0)}{1-\mathcal{E}_D(f_0)}}} - \sqrt{\frac{\tilde{\delta}+1}{\tilde{\delta}}}\right] \cdot \sqrt{\log\left(\frac{4}{\delta}\right)}, \tag{42}$$

$\square$

**Sample Complexity Estimation**  In this paragraph, we present the proof for the sample complexity estimation.

**Theorem A.3.** *Given a function $f_1$ trained by the $N - (\varepsilon, \tilde{\delta}, b)$ training recipe, we have, with at least $(1-\delta)^2$ probability,*

$$p \triangleq \frac{\mathcal{E}_D(f_1) - \mathcal{E}_D^*}{\mathcal{E}_D(f_0) - \mathcal{E}_D^*} \leq p^* \tag{43}$$

*when*

$$N \geq \left(\frac{ak}{p^*(\mathcal{E}_D(f_0) - \mathcal{E}_D^*)}\right)^2 \left[\sqrt{\frac{\tilde{\delta}+1}{\tilde{\delta}-\frac{\mathcal{E}_D(f_0)}{1-\mathcal{E}_D(f_0)}}} - \sqrt{\frac{\tilde{\delta}+1}{\tilde{\delta}}}\right]^2 \cdot \log\left(\frac{4}{\delta}\right) \tag{44}$$

*and*

$$\mathcal{E}_D(f_0) \leq \frac{\tilde{\delta}}{1+\tilde{\delta}} \tag{45}$$

Theorem A.3 specifies the conditions on the number of unlabeled samples, $N$, necessary to ensure that the convergence rate stays below the target rate, $p^*$.

We start by introducing the following lemma:

**Lemma A.4.** *Given a function $f_1$ trained with the $N - (\varepsilon, \tilde{\delta}, b)$ training recipe, and positive constants $c_1$ and $c_2$ such that $\mathcal{E}_D^* + c_1 \leq \mathcal{E}_D(f_0) \leq \mathcal{E}_D^* + c_2$. If $N$ satisfies*

$$N \geq \left(\frac{ak}{p^* c_1}\right)^2 \left[\sqrt{\frac{\tilde{\delta}+1}{\tilde{\delta} - \frac{\mathcal{E}_D^*+c_2}{1-\mathcal{E}_D^*-c_2}}} - \sqrt{\frac{\tilde{\delta}+1}{\tilde{\delta}}}\right]^2 \log\left(\frac{4}{\delta}\right) \tag{46}$$

*and*

$$\mathcal{E}_D(f_0) \leq \frac{\tilde{\delta}}{1+\tilde{\delta}} \tag{47}$$

*then, with at least $(1-\delta)^2$ probability, we have*

$$p \triangleq \frac{\mathcal{E}_D(f_1) - \mathcal{E}_D^*}{\mathcal{E}_D(f_0) - \mathcal{E}_D^*} \leq p^* \tag{48}$$

*Proof.* According to Equations 35 and 36, for a function $f_1$ trained with the $N - (\varepsilon, \tilde{\delta}, b)$ training recipe, we have with at least $1 - \delta$ probability,

$$\mathcal{E}_D(f_1) \leq (1+b)\varepsilon + ak\sqrt{\log\left(\frac{4}{\delta}\right)}\sqrt{\frac{(1+\tilde{\delta})(1-\mathcal{E}_D(f_0))}{\tilde{\delta}(1-\mathcal{E}_D(f_0)) - \mathcal{E}_D(f_0)}}\frac{1}{\sqrt{N}} \triangleq upper(f_1) \tag{49}$$

From this, we can derive:

$$p \triangleq \frac{\mathcal{E}_D(f_1) - \mathcal{E}_D^*}{\mathcal{E}_D(f_0) - \mathcal{E}_D^*} \leq \frac{upper(f_1) - \mathcal{E}_D^*}{\mathcal{E}_D(f_0) - \mathcal{E}_D^*} \triangleq \tilde{p} \tag{50}$$

Therefore, to ensure that inequality 48 holds, the following condition must be satisfied:

$$\tilde{p} \leq p^* \tag{51}$$

From equation 50, we can further derive:

$$upper(f_1) - \mathcal{E}_D^* \leq p^*(\mathcal{E}_D(f_0) - \mathcal{E}_D^*) \tag{52}$$

Substituting equation 35 into equationeq:7 and simplifying yields:

$$\sqrt{\frac{(1+\tilde{\delta})(1-\mathcal{E}_D(f_0))}{\tilde{\delta}(1-\mathcal{E}_D(f_0)) - \mathcal{E}_D(f_0)}}\frac{1}{\sqrt{N}} \leq p^* \frac{\mathcal{E}_D(f_0) - \mathcal{E}_D^*}{ak\sqrt{\log\left(\frac{4}{\delta}\right)}} \tag{53}$$

Rearranging equation 52 with $\mathcal{E}_D^* + c_1 \leq \mathcal{E}_D(f_0) \leq \mathcal{E}_D^* + c_2$ gives us the following inequality about $N$:

$$N \geq \left(\frac{ak}{p^* c_1}\right)^2 \left[\sqrt{\frac{\tilde{\delta}+1}{\tilde{\delta} - \frac{\mathcal{E}_D^*+c_2}{1-\mathcal{E}_D^*-c_2}}} - \sqrt{\frac{\tilde{\delta}+1}{\tilde{\delta}}}\right]^2 \log\left(\frac{4}{\delta}\right) \tag{54}$$

This completes the proof of Lemma A.4. □

Finally, by choosing $c_1 = c_2 = \mathcal{E}_D(f_0) - \mathcal{E}_D^*$ in Lemma A.4, we obtain Theorem A.3.

## A.1 TRAINING RECIPE DETAIL

Our implementation utilizes a ResNet architecture, which is structured with four ResNet blocks. Each block comprises a pair of convolutional layers with 3x3 kernels, as depicted in Figure 4. The model is fine-tuned using a mini-batch size of 128, spanning a total of 200 epochs. The learning rate and momentum are set at 0.1 and 0.9, respectively. Weight decay is regulated at a rate of $1 \times 10^{-4}$. Training checkpoints are stored every 10 epochs. Data loading is executed with four workers, and the model's performance is assessed on a validation set.

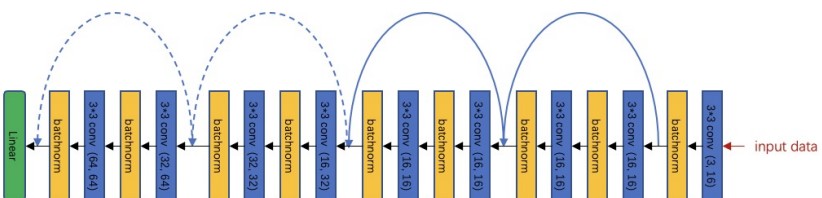

Figure 4: Model architecture

