# OpenReview forum: "Theoretical insights into pseudo-label-based semi-supervised learning: Convergence rate and sample complexity analysis"
_ICLR.cc/2024/Conference — ICLR 2024 Conference Withdrawn Submission_

### Official Review · Reviewer_QKV9 · 2023-10-18

**Soundness:** 2 fair
**Presentation:** 3 good
**Contribution:** 2 fair
**Rating:** 5
**Confidence:** 4

**Summary:**

This paper works on the theoretical analysis of pseudo-label-based approaches for semi-supervised learning (SSL). Based on the theoretical analysis parts of Garg et al., 2021, the paper proposes the optimal population error analysis, convergence rate estimation, and sampling complexity analysis for pseudo-label-based SSL approaches. Experimental results validate the proposed convergence rate estimation results.

**Strengths:**

- The paper is generally well written.
- The problem studied is very important for the literature.
- The theoretical results proposed are new to the literature.

**Weaknesses:**

- The paper needs to make strong assumptions about the model. It needs the model to be underparameterized or have strong regularization techniques and not fit the random labels of the unlabeled data at all. However, current SSL methods are mainly based on DNN, and I think they can easily fit all random labels. For example, the FixMatch approach is based on deep networks, and I think it can fit random labels very well. So I think the application scenario might be limited.

- It has to assign random labels to pseudo-labeled data that is incorrectly predicted by the model, and that is not a common strategy of SSL methods. So it may not accurately reveal the characteristics of SOTA SSL approaches.

**Questions:**

- Is the theoretical analysis really useful for pseudo-label based SSL approaches?
- Are the proposed results applicable to SSL methods based on very deep neural networks?

---

### Official Review · Reviewer_xc7D · 2023-10-29

**Soundness:** 2 fair
**Presentation:** 1 poor
**Contribution:** 1 poor
**Rating:** 3
**Confidence:** 4

**Summary:**

The paper provide theoretical results on pseudo-labeling in semi-supervised learning.

**Strengths:**

The paper provided some theoretical analyses about pseudo-labeling in semi-supervised learning.

**Weaknesses:**

Weaknesses:

Technical:

1. The authors should provide a more detailed explanation of the distinctions between their work and the studies cited as [1] and [2].
2. One of the primary concerns in semi-supervised learning is the limited availability of correctly labeled samples compared to the abundance of unlabeled data. However, this work's reliance on the condition $ \frac{m}{n} < 1 $, as outlined in Definition 3.1, appears to deviate from practical scenarios where $ \frac{m}{n} $ might be significantly larger.
3. The paper references "optimal population risk" without clearly defining the term, which can be confusing.
4. Several key assumptions are not explicitly stated in the paper. For instance, the assumption that the loss function falls within the $[0,1]$ range and, as a consequence, the population risk also lies in that range. Furthermore, the paper assumes that $ (1 - \mathcal{E}_D(f_0))N $ serves as an approximation for the number of correctly labeled samples, but this is not clearly articulated.
5. In Theorem 4.5, when dealing with an infinite number of unlabeled data samples, the condition $ \frac{m}{n} < 1 $ for $ N-(\epsilon, \delta, b) $ is discussed. However, it remains unclear whether Theorem 4.5 considers the case where $ n $ goes to infinity, $ m $ goes to infinity, or both.

Presentation:

1. The paper introduces the variable $ N $ in the abstract without providing a clear definition or explanation.
2. The writing in the introduction section is somewhat unclear. In Paragraph 3 of the introduction, the authors mention "convergence rate" without specifying whether it refers to the convergence rate of an upper bound or the convergence rate in optimization. Clarification is needed.
3. The problem formulation is not presented clearly, which can make it challenging for readers to understand the paper's core concepts.

References:

[1]: He, H., Aminian, G., Bu, Y., Rodrigues, M., & Tan, V. Y. (2023, April). "How Does Pseudo-Labeling Affect the Generalization Error of the Semi-Supervised Gibbs Algorithm?". In International Conference on Artificial Intelligence and Statistics (pp. 8494-8520). PMLR.
[2]: Aminian, G., Abroshan, M., Khalili, M. M., Toni, L., & Rodrigues, M. (2022, May). "An information-theoretical approach to semi-supervised learning under covariate-shift." In International Conference on Artificial Intelligence and Statistics (pp. 7433-7449). PMLR.

**Questions:**

see weaknesses part.

---

### Official Review · Reviewer_71fq · 2023-11-01

**Soundness:** 1 poor
**Presentation:** 2 fair
**Contribution:** 1 poor
**Rating:** 1
**Confidence:** 3

**Summary:**

This article attempted to investigate the performance of semi-supervised learning with pseudo-labels under a general setting. The proposed analysis applies to *training recipes* on correctly labeled and randomly labeled data, defined by three parameters dictating the upper bound of the proportion of randomly labeled data, the upper bound of the empirical error on the correctly labeled data set and the lower bound of the empirical error on the randomly labeled data set. The key contribution is a convergence rate on the updated population error rate given by the current learning model trained with pseudo-labels generated from the previous learning model.

**Strengths:**

This work is well motivated. The use of pseudo-labels to improve learning performance is an interesting and relevant question in machine learning research. In contrast with its empirical success, its theoretical understanding remains to be better developed, specially in general settings.

**Weaknesses:**

My main concern about this work is its technical soundness. Theorem 4.3, which is the key contribution of this work, is built upon Proposition 4.1, which states a result drawn from a previous work (Garg et al., 2021), As Proposition 4.1, this theorem applies to training recipes using a set of correctly labeled data and a set of randomly labeled data. This setting is obviously ill-suited to pseudo-labels, which can not be divided into a set of correct labels and a set of random labels. To remedy this, the authors proposed to select some pseudo-labels and re-randomize them, which seems to me problematic. In Equation (32), the authors seemed to replace the ratio of randomly labeled data over correctly labeled data with the ratio of re-randomized pseudo-labels plus incorrect pseudo-labels over correct pseudo-labels, without proper justification. If the authors could clarify this point, I would be willing to reconsider my score.

**Questions:**

See Weaknesses.

---

### Official Review · Reviewer_XJhG · 2023-11-07

**Soundness:** 1 poor
**Presentation:** 1 poor
**Contribution:** 1 poor
**Rating:** 3
**Confidence:** 5

**Summary:**

The paper studies the problem of semi-supervised learning. In this setting, we have access to a set of unlabeled data, and a set of labeled data, which are both used during learning.  The paper focused on the pseudo-label setting, where the unlabeled data is annotated with "pseudo-labels" that are generated by a previous model.

The paper provides a theoretical study of this setting, and it claims to provide results for the convergence rate and the sample complexity for those problems.

The authors run experiments to corroborate their theoretical findings.

**Strengths:**

The paper studies an important problem. It is true that there is little theoretical understanding of semi-supervised learning, and this is a rich and active area of research.

**Weaknesses:**

In my opinion, the paper is poorly written, it contains several mistakes, and it does not provide novel contributions. In the current state, I cannot recommend for acceptance.

First of all, the theoretical contribution of the paper does not contain any original idea in my opinion. In particular, the definition 3.1 of "training recipe" is simply used to re-write the upper bound of previous work (Proposition 4.1 is Theorem 3 from Garg et al. 2021, also note that we need to specify that we obtain \hat{f} using regularized ERM for those results, since that's when Theorem 3 applies from that work). All the following results are simple corollary or re-arrangements of this upper bound with doubt usefulness (Theorem 4.3, Theorem 4.4, Theorem 4.5).

The definition of training recipe hides most of the important contribution of the original work of Garg et al 2021, and it also hides technical details that the authors do not cover.

In the training recipe, $m/n < \tilde{\delta} <1$ represents the ratio between $m$ unlabeled data and $n$ labeled data, where $N = m+n$ ( this notation is used for the Proposition 4.1 from the previous work on which all the other results are based on)

In the following pages, the definition of $N$ changes: in top of page (6), $N$ becomes the number of unlabeled data. The authors say "this implies that the greater the quantity of unlabeled data available, the quicker our algorithm converges to a solution". This is very misleading, since if $N$ increases, the number of labeled data and unlabeled data both increase (as we need to keep the ratio $\tilde{\delta} <1$).  This mistake is repeated in all the following sections. In section 4.4, they study the case of infinite unlabeled data, but in this case because of the constraint $\tilde{\delta}<1$ we also have an infinite number of labeled data, hence we are not in a semi-supervised setting anymore. (Also, it is not surprising that if the number of labeled data and unlabeled data goes to infinity, we can solve the learning problem)

Aside from this and other technical details, the paper is not formal in its mathematical language. In particular, it uses terms without defining them as sample complexity, and convergence rate, and it seems that their definition is different from the canonical sense.  A few of many examples:

(1) In page 4, they say that the  denote the optimal population error on $\hat{f}$ is $\epsilon_D^*$. Then they claim that they can use Proposition 4.1 to estimate this value. However,  $\epsilon_D^*$ is not defined, and it seems to be simply defined as the minimum value of the upper bound of Proposition 4.1 in the case that we only have labeled data. Even in that case, it is not clear why we need to use Proposition 4.1 if we only use $N$ labeled data points (i.e., m=0), as standard statistical learning tools would suffice (also it seems that Definition 3.1 would not even hold, because there is no set $\tilde{S}$ or we do not have a portion of data that is randomly labeled.
Besides, minimizing an upper bound usually does not provide an "optimal value" unless this upper bound is tight.

(2) The paper talks about convergence rate for the whole paper as the main strength. It is only defined finally in equation (5) at page 5 as an artifact of the computation in the papers, and it is not even clear why the "convergence rate" term is used (what is the sequence?).

(3) Another example of informality is definition 3.1 itself. They say that if (1) holds, "then we can procure a model \hat{f} ... $ and it is not clear whether this is assumption (I think that's the case) or a consequence.

(4) Remark 4.2 is very confusing. Yes, Monte-carlo estimation has error  rate O(1/\sqrt{n}), but what does this say about optimality (you need a lower bound), and how this even relates to definition 3.1?

----

The coverage of the previous work is very weak and unclear. In section 2.2, the paper only mentions the method of using the complexity of hypothesis class for population error of DNN model, that is known to give vacuous upper bound for those architectures.
Additionally, all the results of this paper are grounded on the results of Garg et al., 2021, and they simply state very vaguely "A new perspective in the field was brought forth by (Garg et al, 2021)".

----


Minor remark: (1-\delta)^2 looks not natural to me. I looked at the appendix to see how  this is obtained, but I could not find it. By taking an union bound, you should obtain something that still looks like 1-\delta

The experiments of figure 1 need multiple runs with standard deviation. It is not clear how much the results are affected by the initialization, as they all seem to converge to the same result when the number of epochs increases

----


Other minor things:
"equationeq:7" page 15,
"under-parametrized or when applied" ->"under-parametrized models" in page 1

**Questions:**

What are the original ideas and contributions of this paper with respect to the work of Garg et al 2021? What technical novelty do you introduce in proving your results?

If the number of unlabeled data goes to infinity, then the number of labeled data also needs to go to infinity to preserve the ratio $\tilde{\delta}<1$. Why is this setting interesting in that case, and how does this affect the conclusion of section 4.2, 4.3 and 4.4

For the "optimal population error", if all the data is correctly labeled, what is the point of the parameter b in equation 7?